# Beyond Detection: Comparative Explainability Study on Trypanosoma cruzi Using CAMs and DETR Attention

Aqsa Yousaf*, Paul Agbaje*, Megan Coffee†, Habeeb Olufowobi*

*Department of Computer Science and Engineering, University of Texas at Arlington, Arlington, TX, USA
†Department of Medicine, Division of Infectious Diseases, NYU Grossman School of Medicine, New York, NY, USA
{aqsa.yousaf, pauloluwatowoju.agbaje, habeeb.olufowobi}@uta.edu, Megan.coffee@nyulangone.org

*Abstract*—Chagas disease, caused by Trypanosoma cruzi, demands accurate and interpretable detection methods to support clinical decision-making. While deep learning models such as YOLOv8 and DINO-DETR perform well on microscopy images, their lack of interpretability hinders clinical adoption. We present the first comparative explainability study of CNN- and transformer-based object detectors for Trypanosoma cruzi detection. For YOLOv8, we benchmark ten Class Activation Mapping explainable AI (CAM-XAI) methods across multiple internal layers, evaluating interpretability using Intersection-over-Union (IoU) and Energy-Based Pointing Game (EBPG). For DINO-DETR, we introduce a query-specific attention visualization method that maps decoder attention of a query to image space without backpropagation. Our results reveal complementary behaviors: CAMs highlight broad parasite regions, while DETR attention targets fine-grained, discriminative features. We further demonstrate that existing localization metrics are inadequate for shared heatmaps in multi-object settings, underscoring the need for new localization evaluation metrics in medical explainability.

*Index Terms*—Explainable AI, Chagas disease, Trypanosoma cruzi, Class Activation Mapping, Medical image analysis, Blood parasites, Microscopy

## I. INTRODUCTION

Chagas disease is a life-threatening parasitic illness caused by the protozoan *Trypanosoma cruzi* (*T. cruzi*). It is classified by the World Health Organization (WHO) as a neglected tropical disease, endemic to 21 countries in the Americas [1], affecting approximately 7 million people and causing an estimated 10,000 deaths annually [2]. The disease progresses through two phases: acute and chronic. The acute phase typically lasts around two months, during which the parasite is abundant in the bloodstream. Early detection and treatment in this phase are critical to preventing progression to the chronic stage [2].

Traditional diagnosis through microscopy requires highly trained personnel and specialized equipment [3], such as professional-grade microscopes with high-resolution imaging capabilities. This process is labor-intensive, time-consuming, and often inaccessible in resource-limited settings. These limitations contribute to diagnostic delays and underdiagnosis

in endemic regions, emphasizing the need for automated and cost-effective methods to detect parasites.

Machine learning (ML) and computer vision techniques have emerged as promising solutions for parasite detection in microscopy images [4–6]. Recent work has demonstrated strong detection performance across various architectures. For instance, YOLO-based frameworks [4], vision transformers [7], detection transformer (DETR-based models) [8], and hybrids such as MeDINO [9] have been explored. Prior work also demonstrates the viability of low-cost imaging (e.g., smartphone microscopy) using traditional ML classifiers [5], laying the foundation for deep learning (DL) systems that can directly learn parasite features from raw images.

By automating the detection of *T. cruzi* trypomastigotes in blood smear images, ML-based approaches can improve diagnostic speed and consistency. However, despite gains in detection accuracy, ML models often lack transparency, limiting trust and adoption in clinical workflows [10]. Interpretability is essential to ensure that model predictions are grounded in biologically meaningful features rather than spurious correlations or background artifacts. This is particularly critical in clinical settings, where clinician trust depends not only on correctness but on the transparency of the decision-making process.

Explainable AI (XAI) methods aim to address this challenge by generating human-interpretable explanations of model predictions. In medical imaging, XAI can highlight regions of interest that influenced the model's output, enabling practitioners to verify that the model is focusing on clinically meaningful structures (e.g., parasite morphology rather than irrelevant artifacts). Despite growing interest in XAI, prior work on *T. cruzi* detection has largely prioritized performance benchmarks, with limited attention to interpretability. This gap in model transparency remains a key barrier to clinical deployment.

In this paper, we bridge this gap by performing a comparative explainability study of convolutional neural network (CNN) and transformer-based object detection models for automated parasite detection. Our contributions are as follows:

- We introduce a novel query-specific attention visualization method for DINO-DETR that extracts decoder attention weights and corresponding spatial sampling

The research reported in this paper was supported by AIM-AHEAD Coordinating Center, award number OTA-21-017, and was, in part, funded by the National Institutes of Health Agreement No. 1OT2OD032581.

locations, producing fine-grained, interpretable saliency maps.

- We conduct the first comparative analysis of explainability techniques across CNN-based (YOLOv8) and transformer-based (DINO-DETR) detection architectures using quantitative localization metrics.
- We present the first systematic benchmarking of ten gradient-based CAM-XAI methods across multiple YOLOv8 internal layers, identifying optimal method-layer configurations for visualizing *T. cruzi* features in blood smear images.

## II. BACKGROUND AND RELATED WORK

DL has transformed medical image analysis, enabling the development of automated tools for disease diagnosis and cell detection. For parasitic infections like Chagas disease, object detection models offer a promising solution by identifying *T. cruzi* trypomastigotes in blood smear images, supporting faster diagnosis in both clinical and resource-limited settings. CNNs and transformer-based detectors have emerged as dominant architectural paradigms for these tasks. However, limited interpretability remains a critical barrier to clinical integration. This section reviews prior work on parasite detection and XAI, with a focus on their application to *T. cruzi* detection in blood smear images.

### A. Parasite Detection in Blood Smear Images

Early approaches to parasite detection relied on classical ML methods. For example, Morais et al. [5] used a random forest classifier on handcrafted morphological and textural features extracted from smartphone-acquired images, achieving an 89% F1 score. While this work demonstrated the feasibility of mobile microscopy, the approach required manual feature engineering and lacked the scalability and generalizability of deep learning models.

More recently, DL-based object detectors have become state-of-the-art. Mura et al. [4] proposed YOLO-Tryppa, a YOLO-based model evaluated on *T. brucei* images, achieving a mean average precision (mAP$_{50}$) of 71.3%. Rada et al. [6] evaluated multiple object detectors, including RetinaNet, Faster R-CNN, FCOS, and YOLOv8, on *T. cruzi* microscopy images. YOLOv8 achieved the highest performance (mAP$_{50}$ = 0.951), even on low-resolution clinical images, which shows its robustness and suitability for real-world deployment in resource-constrained environments.

Transformer-based object detectors are emerging as an alternative to CNNs. DETR and its variants, such as DINO-DETR, utilize encoder-decoder attention mechanisms to detect objects without requiring anchor boxes or region proposals. Guemas et al. [11] and Nakarmi et al. [12] explored the combination of Deformable DETR and CNNs for parasite detection. Lin et al. [7] compared ViT and YOLOv8 for parasite classification, while Miao et al. [9] introduced MeDINO, a DINO-DETR variant optimized for medical imaging tasks. However, to our knowledge, no prior work has applied DETR-based models

to *T. cruzi* detection specifically, nor has any evaluated their interpretability in this context.

### B. Explainable AI for Parasite Detection

Despite impressive performance, most existing parasite detection studies emphasize detection accuracy, with limited attention to how or why predictions are made. In medical imaging, where diagnostic decisions directly impact patient outcomes, this opacity undermines trust and slows clinical adoption. In high-stakes applications, such as parasite detection, false positives or negatives can lead to misdiagnosis, making interpretability crucial. XAI aims to bridge this gap by revealing which image regions or features influence a model's decision, helping clinicians verify that predictions align with biological and diagnostic expectations.

Among XAI methods, class activation mapping (CAM) techniques are widely used for visualizing model decisions. Grad-CAM [13] is a foundational gradient-based method that highlights important regions in intermediate feature maps. Its variants improve localization in various ways: Grad-CAM++ [14] uses higher-order gradients for small or overlapping objects, while XGrad-CAM [15] and LayerCAM [16] provide fine-grained localization using axiomatic or pixel-wise mechanisms. Gradient-free methods, such as EigenCAM [17], utilize the principal components of feature activations to generate class-agnostic saliency maps, thereby enhancing robustness to model noise. EigenGradCAM [18] combines gradient-based localization with dimensionality reduction for smoother visualizations. HiResCAM [19] avoids global pooling to improve resolution, while KPCA-CAM [20] captures nonlinear patterns in feature space. RandomCAM [18] serves as a baseline sanity check. These methods differ in localization precision, computational cost, and robustness to noise. While commonly applied to classification tasks such as tumor detection and pathology analysis, the systematic evaluation of these methods in parasite detection remains limited.

With the advent of transformer-based detectors, attention maps have emerged as a complementary form of explainability. DETR models compute deformable attention weights that connect object queries to image regions, providing insight into the model's internal reasoning. Visualizing decoder or self-attention maps can highlight which regions the model focuses on when generating predictions. Unlike CAMs, these maps are intrinsic to the model's forward pass. Several studies suggest that attention-based explanations can correlate well with semantically meaningful regions and, under certain conditions, outperform CAMs in highlighting objects of interest [21, 22].

To our knowledge, no prior work systematically evaluates the interpretability of DETR-based and CNN-based detectors for *T. cruzi*. This gap motivates our work, which provides a comprehensive comparison of CAM-based and attention-based XAI methods for parasite detection in blood smear microscopy.

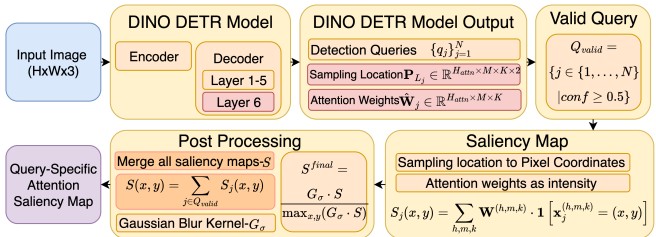

Fig. 1. Overview of the deformable attention-based explainability pipeline. An input image is processed by DINO-DETR, generating queries $\{q_j\}_{j=1}^N$. From the 6th decoder layer, query-specific sampling locations $\mathbf{P}_{L_j} \in \mathbb{R}^{H_{attn} \times M \times K \times 2}$ and attention weights $\hat{\mathbf{W}}_j \in \mathbb{R}^{H_{attn} \times M \times K}$ are extracted. Queries with confidence scores greater than 0.5 are used to generate saliency maps $S_j(x, y)$, where attention weights define intensity values at sampled locations. Final map $S^{final}$ is obtained by aggregating and smoothing individual maps using Gaussian blur.

## III. METHODOLOGY

### A. Dataset

We use the publicly available dataset of blood smear images, introduced by Morais et al. [5], which consists of 674 high-resolution microscopy images (3456×4608 pixels) of 1,314 *T. cruzi* parasites' instances, captured using a smartphone camera attached to an optical microscope. The dataset was acquired from Giemsa-stained blood smears prepared from Swiss mice infected with *T. cruzi* during the acute phase of infection. We randomly selected 569 images for training and 105 for validation.

### B. Parasite Detection Models

We evaluated two representative object detection architectures: YOLOv8, a state-of-the-art CNN-based model, and DINO-DETR, a representative transformer-based baseline.

We selected YOLOv8 due to its state-of-the-art performance on medical datasets and its proven accuracy in related work. Rada et al. [6] and Mura et al. [4] benchmarked several YOLO object detectors on *T. cruzi* images and found YOLOv8 outperformed alternatives like RetinaNet and FCOS, achieving high mAP scores even under constrained imaging conditions.

To represent the transformer-based detection family, we adopted DINO-DETR as a strong baseline model. It has been effectively used in prior medical imaging studies such as MeDINO [9], and it retains the original encoder-decoder attention architecture of DETR, which is essential for generating query-based visual explanations. Moreover, since DINO-DETR follows the general DETR design, our explainability approach is broadly applicable to other DETR-family models, making it a representative and extensible choice for transformer-based explainability analysis.

TABLE I
PERFORMANCE OF YOLOv8 AND DINO-DETR ON *T. cruzi* DETECTION

| Model | Prec. | Rec. | mAP @0.5 | mAP @0.5:0.95 | AR 100 | Inference Time (ms) |
|---|---|---|---|---|---|---|
| YOLOv8 | 0.864 | 0.903 | **0.913** | **0.432** | – | **12.8** |
| DINO-DETR | – | – | 0.872 | 0.420 | 0.599 | 29.9 |

### C. CAM-Based Explainability for YOLOv8

To interpret YOLOv8's predictions, we selected CAM-based methods due to their low inference time and ability to produce spatially informative explanations, making them well-suited for time-sensitive clinical applications [23]. We adapted ten CAM methods: Grad-CAM, Grad-CAM++, EigenCAM, EigenGradCAM, XGradCAM, LayerCAM, HiResCAM, RandomCAM, KPCA-CAM, and GradCAM-ElementWise (GradCAM-EW). While these techniques were originally developed for image classification, we modified them to support object detection by aggregating anchor-level predictions into a unified score per image, which was then passed to the CAM computation pipeline. We extended the PyTorch-Grad-CAM library to accommodate YOLOv8's multi-scale detection heads. The resulting saliency maps reveal spatial regions that most influence the model's predictions, aiding both qualitative and quantitative analysis of interpretability.

### D. Deformable Attention-Based Explainability for DINO-DETR

In transformer-based detectors like DINO-DETR, each object prediction is generated from a distinct decoder query. This structure makes query-specific attention well-suited for producing interpretable explanations that maintain a clear correspondence between individual predictions and the regions they attend to. Existing explainability methods for transformer detectors are typically not query-specific or rely on complex post-hoc computations. To address this, we introduce a simple, architecture-aligned approach that extracts decoder-level deformable attention weights and sampling locations to generate spatially precise, query-specific saliency maps. This method avoids backpropagation and enables localized, per-object explanations that complement CAM-based methods and highlight cases where convolutional saliency may be diffuse or ambiguous.

### E. Deformable Attention-Based Saliency Extraction

**Prediction and Query Selection.** Given an input image $\mathbf{I} \in \mathbb{R}^{H \times W \times 3}$, DINO-DETR produces $N$ decoder queries $\{q_j\}_{j=1}^N$ with confidence scores $s_j$. We retain high-confidence queries:

$$\mathcal{Q}_{\text{valid}} = \{j \in \{1, \dots, N\} \mid s_j \geq 0.5\} \quad (1)$$

**Deformable Attention Extraction.** For each valid query $q_j$, we extract sampling locations: $\mathbf{P}_j \in \mathbb{R}^{H_{\text{attn}} \times M \times K \times 2}$ and Attention weights of last decoder layer: $\mathbf{W}_j \in \mathbb{R}^{H_{\text{attn}} \times M \times K}$, where $H_{\text{attn}}$ is the number of attention heads, $M$ is the number of feature pyramid levels, and $K$ is the number of sampling points per level.

**Coordinate Conversion.** The normalized sampling locations are converted to discrete pixel coordinates:

$$\mathbf{x}_j^{(h,m,k)} = \left( \text{round}(P_j^{(h,m,k)}[0] \times W), \text{round}(P_j^{(h,m,k)}[1] \times H) \right) \quad (2)$$

where $W$ and $H$ are the image width and height respectively, and round($\cdot$) denotes rounding to the nearest integer.

**Per-Query Saliency Map Construction.** We construct a query-specific saliency map by accumulating attention weights at their corresponding pixel locations:

$$\mathbf{S}_j(x,y) = \sum_{h,m,k} \mathbf{W}_j^{(h,m,k)} \cdot \mathbf{1}\left[\mathbf{x}_j^{(h,m,k)} = (x,y)\right] \quad (3)$$

where $\mathbf{1}[\cdot]$ is the indicator function that equals 1 when the condition is true and 0 otherwise.

**Aggregation Across Queries.** To obtain a unified saliency map for the image, we aggregate contributions from all valid queries:

$$\mathbf{S}(x,y) = \sum_{j \in \mathcal{Q}_{\text{valid}}} \mathbf{S}_j(x,y) \quad (4)$$

**Post-processing.** The aggregated saliency map is smoothed using a Gaussian kernel $G_\sigma$ with standard deviation $\sigma$ and normalized to the range $[0,1]$:

$$\mathbf{S}^{\text{final}} = \frac{G_\sigma * \mathbf{S}}{\max_{x,y}(G_\sigma * \mathbf{S})} \quad (5)$$

where $*$ denotes convolution.

**Interpretability.** The final saliency map $\mathbf{S}^{\text{final}}$ highlights spatial regions most influential to the model's predictions. By leveraging decoder deformable attention without requiring gradients, this method provides faithful, architecture-aware explanations suited to transformer-based object detectors.

### F. Evaluation Metrics for XAI Evaluation

To evaluate the spatial alignment between saliency maps and ground truth objects in object detection, we consider two widely adopted localization metrics, EBPG, and IoU.

**Energy-Based Pointing Game.** EBPG measures the proportion of total saliency energy that lies inside the ground truth regions. Let $\mathcal{S}(p)$ denote the saliency at pixel $p$ and $\Omega_{\text{GT}}$ the union of all ground truth bounding box regions. The EBPG score is given by:

$$\text{EBPG} = \frac{\sum_{p \in \Omega_{\text{GT}}} \mathcal{S}(p)}{\sum_p \mathcal{S}(p)} \quad (6)$$

**Intersection over Union.** IoU compares a thresholded saliency map (converted to a binary mask) to the ground truth regions. Let $\mathcal{T}$ be a threshold applied to the saliency map, and let $\mathcal{S}_{\text{bin}}$ be the resulting binary map. The IoU is defined as:

$$\text{IoU} = \frac{|\mathcal{S}_{\text{bin}} \cap \Omega_{\text{GT}}|}{|\mathcal{S}_{\text{bin}} \cup \Omega_{\text{GT}}|} \quad (7)$$

### G. Implementation Details

Our experiments were conducted on an NVIDIA RTX 6000 Ada Generation GPU with 48 GB of VRAM. We trained the YOLOv8m and DINO-DETR (4-scale ResNet backbone) object detection models for *T. cruzi* detection using their default hyperparameter settings and 25 epochs. To binarize the saliency maps, we applied a fixed threshold of $\tau = 0.3$. We used a Gaussian kernel size of $(11 \times 11)$ to smooth the saliency maps, where the standard deviation $\sigma$ was automatically computed by OpenCV based on the kernel size.

## IV. RESULTS

### A. Parasite Detection

Table I presents the detection performance metrics on validation dataset. YOLOv8, achieved superior detection performance with an $\text{mAP}_{0.5}$ of 0.913, precision of 0.864, and recall of 0.903. DINO-DETR achieved an $\text{mAP}_{0.5}$ of 0.872 and an Average Recall at 100 detections ($\text{AR}_{100}$) of 0.599. The higher mAP of YOLOv8 indicates its effectiveness in accurately localizing *Trypanosoma cruzi* parasites in microscopic blood smear images.

### B. Explainability Analysis of YOLOv8 using Class Activation Mapping (CAM) Methods

To assess the interpretability of YOLOv8 for parasite detection, we applied ten Class Activation Mapping (CAM) techniques on the validation set using the fine-tuned YOLOv8 model and measured their performance using IoU and EBPG metrics. These metrics capture region-level overlap and fine-grained saliency accuracy, respectively.

**Observation 1: The choice of layer substantially impacts the quality of CAM explanations.** We observe that the effectiveness of each CAM method is highly dependent on the specific layer from which it is computed. Tables II and III show that performance varies considerably across layers (12, 15, 17, 21), with most methods achieving their best results at Layer 17. For instance, *GradCAM++* and *EigenGradCAM* reach peak IoU scores of 0.1482 and 0.1303, and EBPG scores of 0.1220 and 0.4375, respectively. Saliency maps at Layer 17 consistently localize parasite regions, suggesting that intermediate features strike a balance between spatial detail and semantic relevance as shown in Figure 2. These findings highlight the importance of carefully selecting the right intermediate layer used for explanation, as suboptimal choices can degrade interpretability.

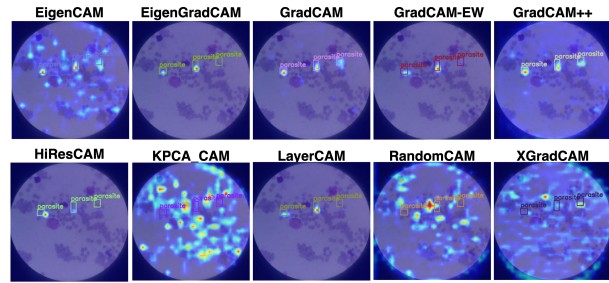

Fig. 2. Visual comparison of CAM methods applied to YOLOv8 outputs using Layer 17, which yielded the best or near-best performance for most methods across both IoU and EBPG metrics. The saliency maps demonstrate that Layer 17 captures consistent and informative visual cues for localizing *T. cruzi* parasites across different CAM approaches. All maps are overlaid on the same predicted image, with bounding boxes corresponding to YOLOv8 model predictions.

**Observation 2: No single layer optimally balances spatial alignment and pixel-level relevance across CAM methods.** Tables II and III show a divergence between IoU and EBPG scores across layers, indicating that region-overlap and point-wise relevance are not simultaneously maximized.

TABLE II
AVG. IoU SCORES ACROSS LAYERS FOR EACH CAM METHOD, EVALUATED ON THE VALIDATION DATASET (HIGHER IS BETTER).

| Layer | EigenGrad | GradCAM | EigenCAM | XGradCAM | RandomCAM | LayerCAM | KPCA-CAM | HiResCAM | GradCAM++ | GradCAM-EW |
|---|---|---|---|---|---|---|---|---|---|---|
| Layer 12 | **0.2712** | 0.0131 | 0.0003 | 0.0081 | 0.0049 | 0.0167 | 0.0000 | **0.3446** | 0.0052 | 0.4310 |
| Layer 15 | 0.1025 | 0.0159 | 0.0006 | 0.0301 | 0.0691 | 0.2763 | 0.0188 | 0.1861 | 0.0304 | **0.4673** |
| Layer 17 | 0.1303 | **0.2912** | **0.1583** | **0.1292** | **0.1135** | 0.1659 | **0.0507** | 0.1183 | **0.1482** | 0.1823 |
| Layer 21 | 0.0000 | 0.0000 | 0.0198 | 0.0000 | 0.0133 | 0.0000 | 0.0000 | 0.0000 | 0.0000 | 0.0000 |
| Layer 15+17 | 0.1846 | 0.0797 | 0.0090 | 0.0574 | 0.0882 | **0.3016** | 0.0313 | 0.1998 | 0.0784 | 0.3957 |
| Layer 15+17+21 | 0.1846 | 0.0797 | 0.0119 | 0.0574 | 0.0239 | 0.3016 | 0.0060 | 0.1998 | 0.0784 | 0.3957 |

TABLE III
AVG. EBPG SCORES ACROSS LAYERS FOR EACH CAM METHOD, EVALUATED ON THE VALIDATION DATASET (HIGHER IS BETTER).

| Layer | EigenGrad | GradCAM | EigenCAM | XGradCAM | RandomCAM | LayerCAM | KPCA-CAM | HiResCAM | GradCAM++ | GradCAM-EW |
|---|---|---|---|---|---|---|---|---|---|---|
| Layer 12 | 0.2550 | 0.0123 | 0.0004 | 0.0096 | 0.0061 | 0.0843 | 0.0001 | 0.3669 | 0.0070 | 0.3489 |
| Layer 15 | 0.4208 | 0.0128 | 0.0013 | 0.0141 | **0.0652** | **0.5988** | 0.0226 | **0.4005** | 0.0209 | **0.4705** |
| Layer 17 | **0.4375** | **0.1749** | **0.0864** | **0.0584** | 0.0612 | 0.3925 | **0.0265** | 0.3172 | **0.1220** | 0.3213 |
| Layer 21 | 0.0000 | 0.0000 | 0.0203 | 0.0000 | 0.0136 | 0.0000 | 0.0003 | 0.0000 | 0.0000 | 0.0000 |
| Layer 15+17 | 0.3777 | 0.0192 | 0.0079 | 0.0193 | 0.0317 | 0.4936 | 0.0243 | 0.3640 | 0.0284 | 0.4083 |
| Layer 15+17+21 | 0.3777 | 0.0192 | 0.0138 | 0.0193 | 0.0166 | 0.4936 | 0.0085 | 0.3640 | 0.0284 | 0.4083 |

IoU measures the spatial overlap with ground-truth bounding boxes, while EBPG assesses whether the most salient point falls within the target region. *LayerCAM*, *RandomCAM*, and *HiResCAM* achieve their best IoU and EBPG scores on different layers. For example, *LayerCAM* performs best in IoU on Layer 15+17 (0.3016), but its highest EBPG score (0.5988) occurs on Layer 15 alone. Likewise, *HiResCAM* achieves peak IoU on Layer 12 (0.3446) and peak EBPG on Layer 15 (0.4005). To further illustrate this inconsistency, Figure 3 presents qualitative visualizations of saliency maps for the same image. The top row displays CAM outputs generated from the layer with the highest IoU, showing broader and more diffuse heatmaps that roughly align with object boundaries. The bottom row, by contrast, displays outputs from the layer with the highest EBPG, where the saliency is more concentrated and localized around key discriminative regions, often at the center of parasite bodies. These inconsistencies highlight that no single layer optimally satisfies both region-overlap and pixel-level relevance criteria.

These findings show that the choice of layer strongly affects explanation quality and that the optimal layer varies by metric, highlighting the need for multi-metric evaluation in medical XAI benchmarking.

### C. Explainability of DINO-DETR using Deformable Attention Maps

To explore explainability in transformer-based object detection, we implemented a query-specific, attention-based visualization method for DINO-DETR. As detailed earlier, we extracted deformable attention weights and their associated sampling locations directly from the last decoder layer of DINO-DETR for high-confidence queries. These weights were used to construct per-query attention maps by mapping the sampled locations (in normalized coordinates) to absolute pixel coordinates, followed by Gaussian smoothing and normalization for visualization.

**Observation 3: DINO-DETR attention localizes compact regions corresponding to parasite centers.** Figure 5 illustrates the resulting attention maps alongside ground truth

annotations and predicted bounding boxes. The attention maps from DINO-DETR tend to concentrate around compact, high-confidence regions, often aligning with parasite centers, unlike CNN-based CAM methods, which typically activate over broader spatial areas. This focused behavior suggests that transformer-based detectors attend more precisely to discriminative visual features, such as nuclei or dense morphological markers, making their outputs more semantically targeted. The contrast in spatial behavior highlights that CNN and transformer architectures learn different visual representations for parasite detection, suggesting potential benefits for ensemble-based interpretability in medical image analysis.

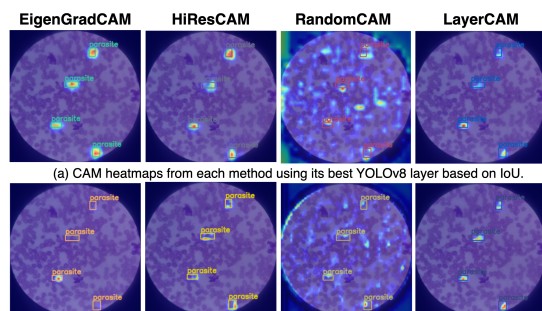

(a) CAM heatmaps from each method using its best YOLOv8 layer based on IoU.

(b) CAM heatmaps from each method using its best YOLOv8 layer based on EBPG.

Fig. 3. Qualitative comparison of CAM methods where the best-performing layer differs across IoU and EBPG metrics. The top row (a) shows saliency maps for each method using the layer with the highest IoU score, while the bottom row (b) uses the layer with the highest EBPG score. This highlights the layer-dependent nature of each evaluation metric.

**Observation 4: DINO-DETR demonstrates strong point-wise localization of discriminative features, despite lower spatial coverage.** Quantitative evaluation shows that DINO-DETR attention maps achieve an average IoU of 0.0949 and EBPG of 0.8063 on the validation dataset. While its saliency maps exhibit limited spatial overlap with annotated parasite regions, they consistently focus on highly informative points. As shown in Table IV, the highest IoU among CAM methods is achieved by GradCAM-EW at Layer 15, whereas LayerCAM yields the highest EBPG. Notably, DINO-DETR surpasses

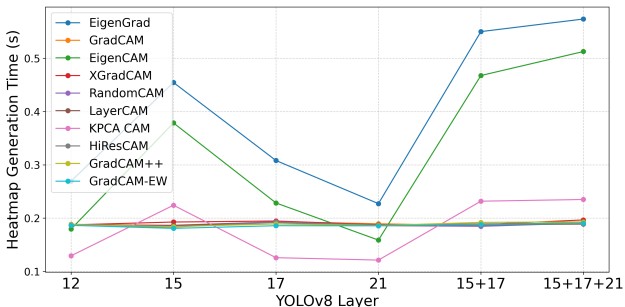

Fig. 4. Average XAI heatmap generation time (in seconds) for various CAM methods across YOLOv8 layers, computed on the validation dataset.

both in EBPG, indicating that its attention mechanism more reliably identifies semantically meaningful regions, even if the spatial extent is coarse. This highlights a trade-off between region-level coverage (IoU) and point-wise semantic precision (EBPG) across architectures.

**Observation 5:** Figure 4 summarizes the average inference time for various CAM methods across YOLOv8 layers. Eigen-GradCAM, EigenCAM, and KPCA-CAM consistently take longer on Layer 15 due to its high spatial resolution (80×80), which significantly increases processing time, especially for eigen-based methods that perform decomposition over large activation maps. The dominance of Layer 15 is also evident in combined layers (e.g., 15+17, 15+17+21), where it drives the overall inference time. Layer 17 shows higher latency than Layer 21 across all methods, despite both having 576 channels, because it operates at a higher spatial resolution (40×40 vs. 20×20), which impacts computation cost.

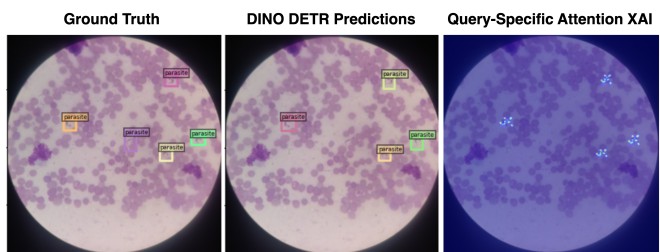

Fig. 5. Comparison of ground truth, predicted detections, and attention-based explanations for a sample blood smear image.

## V. DISCUSSION

**D1: Shared heatmaps align with feature-sharing architectures and clinical expectations.** Our design choice to generate a single explanation heatmap for all the predictions per image reflects the underlying architecture of models such as YOLOv8 and DINO-DETR, which compute predictions from a shared feature representation. A unified heatmap more accurately represents the model's internal reasoning and decision-making process. From a clinical perspective, a single interpretable visualization is more aligned with diagnostic workflows, where the goal is to validate the model's focus across the image holistically, rather than interpreting each prediction in isolation.

TABLE IV
COMPARISON OF LOCALIZATION METRICS FOR TOP CAM METHODS
AND QUERY-SPECIFIC ATTENTION XAI

| Method (Layer) | Avg. IoU | Avg. EBPG |
|---|---|---|
| GradCAM-EW (Layer 15) | **0.4673** | 0.4705 |
| LayerCAM (Layer 15) | 0.2763 | **0.5988** |
| DINO-DETR (Final Decoder Layer, 6[th]) | 0.0949 | **0.8063** |

**D2: Current localization metrics are misaligned with shared heatmap assumptions.** Conventional localization metrics such as IoU, the Pointing Game (PG), and EBPG assume that each object is explained using a dedicated saliency map [24]. When applied to shared explanations, as in our setup, these metrics can yield misleading evaluations. For instance, PG computes whether the most activated pixel lies within a ground truth box, but ignores the overall spatial distribution of saliency. In multi-object settings, this leads to overestimation: a single correct pixel can yield a perfect score even when other targets are missed entirely. EBPG improves on PG by integrating energy within each bounding box. However, it still assigns zero penalty for objects that receive no attention at all, allowing a single well-localized object to dominate the score.

On the other hand, IoU penalizes heatmaps that accurately highlight semantically meaningful subregions, such as the nucleus of a Trypanosoma cruzi parasite, if they do not span the entire ground-truth bounding box.

**D3: A new localization metric is needed for dense predictions per saliency map.** Our findings reveal the inadequacy of current interpretability localization metrics in dense, multi-object settings with a shared saliency map. To address this, we advocate for the development of a new localization metric that accounts for shared saliency attribution.

## VI. LIMITATIONS AND FUTURE WORK

While this study provides valuable insights into the explainability of object detection models for T. cruzi detection, a few areas remain open for further development. First, clinician involvement was not included in the evaluation of saliency maps. Although our visualizations were designed to reflect known morphological features of the parasite, expert feedback would offer stronger validation of their clinical utility. Second, through our analysis, we found that commonly used localization metrics such as PG, IoU, and EBPG are insufficient for evaluating shared saliency maps in dense, multiobject scenarios. This is not only a limitation but also one of the key findings of our study. Finally, this work focuses on a single-parasite dataset. In future research, we plan to expand our analysis to multi-parasite detection and investigate whether explainability methods remain reliable across varying morphological structures and prediction challenges. We also aim to develop new evaluation metrics that address these shortcomings and better reflect the interpretability needs of detection models.

## VII. Conclusion

We present explainability analysis of deep learning models for Trypanosoma cruzi detection in microscopy images, comparing CAM-based methods for YOLOv8 and attention-based explanations for DINO-DETR. For YOLOv8, we benchmarked ten CAM variants across layers using IoU and EBPG. For DINO-DETR, we introduced a query-specific attention visualization method that produces saliency maps without backpropagation. Our findings reveal that CAM methods emphasize broader image regions, while DINO-DETR localizes parasite features with higher semantic precision. We also highlight limitations of existing metrics when evaluating shared heatmaps in multi-object settings. Our findings provide practical insights for model interpretability and underscore the need for localization evaluation metrics in clinical AI applications.

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
