# OpenReview forum: "Beyond Detection: Comparative Explainability Study on Trypanosoma cruzi Using CAMs and DETR Attention"
_IEEE.org/EMBS/BHI/2025/Conference — BHI 2025_

### Official Review · Reviewer_hNQH · 2025-07-13
**XAI for T. cruzi**

**Confidence:** 3
**Clarity Of Writing:** good
**Clinical Significance:** great
**Methodological Novelty:** good
**Overall Rating:** 6

**Experiments And Results:**

good

**Questions For The Authors:**

See previous section

**Strengths:**

1. The choice of YOLO as one of the main models is well-motivated, given its wide adoption in biomedical image analysis tasks, particularly in object detection scenarios.

2. The evaluation metrics: Energy-Based Pointing Game and IoU, are appropriate and effectively support the comparative analysis of the explanation methods. Morever, the paper demonstrates limitation of these metrics.

3. The application domain, while focused on T. cruzi, offers insights that are potentially generalizable to other image-based classification problems in medical imaging.

4. The manuscript is generally well-structured, and the experimental design is clear

**Summary Of The Paper:**

The manuscript presents a comparative study of Class Activation Mapping (CAM) and DETR attention for explainability in the context of T. cruzi image classification. The paper employs YOLO and DETR as the underlying models and use Energy-Based Pointing Game and Intersection over Union (IoU) as evaluation metrics to assess the quality of the explanations generated by these attention mechanisms.

**Weaknesses:**

1.	The manuscript would benefit from an explicit discussion of its limitations.
2.	While the study focuses on model-specific methods, it would be highly valuable to include model-agnostic XAI approaches such as SHAP, or DeepLIFT. These methods are widely used and provide a useful benchmark. A comparison will help clarify the advantages or trade-offs between model-specific and model-agnostic explainability tools, offering deeper insights for the broader community.
3.	The manuscript should be carefully proofread to address minor typos. For example, on Page 3, second paragraph, the phrase “on . cruzi images” should be corrected to “on T. cruzi images.”

---

### Official Review · Reviewer_cMVP · 2025-07-14
**Trypanosoma cruzi  detection**

**Confidence:** 4
**Clarity Of Writing:** excellent
**Clinical Significance:** good
**Methodological Novelty:** great
**Overall Rating:** 7

**Experiments And Results:**

excellent

**Questions For The Authors:**

How to determine the threshold tau for the saliency map in order to calculate IoU defined in equation (7).

If IoU is a desired evaluation criteria, then the need for CAM may be diminished as observer can decide whether to adopt the model output by evaluating the final location/segmentation map directly. Some clarifications on choosing IoU is desired.

**Strengths:**

One strength of the presented work is comprehensive evaluation of different CAM based strategies. The other strength includes detailed explainability analyses and well-articulated observations of the attention maps.

**Summary Of The Paper:**

This work examines a variety of class activation mapping (CAM) based methods for model explainability in YOLOv8 and introduces a query-specific attention visualization scheme for a transformer-based model (DINO-DETER) for Trypanosoma cruzi detection on microscopic images.

**Weaknesses:**

Unlike classification or prediction tasks, explainability may not be a significant limiting factor for clinical adoption in segmentation or detection tasks, as the identified objects (e.g., parasites in this study) can be readily verified by clinicians. As such, the authors may clarify or make a stronger justification for the need for explainability in the specific task in the work in the Introduction section.

---

### Official Review · Reviewer_gCMR · 2025-07-16
**Beyond Detection: Comparative Explainability Study on Trypanosoma cruzi Using CAMs and DETR Attention**

**Confidence:** 2
**Clarity Of Writing:** great
**Clinical Significance:** great
**Methodological Novelty:** great
**Overall Rating:** 6
**Final Rating:** 7

**Experiments And Results:**

great

**Questions For The Authors:**

Can you give more detail on how much data was used to fine-tune each model?
How long does it take to run each model in practice? This is important for real use in hospitals.
Do you plan to test this system in real-time clinical workflows like in radiology departments?
How were the clinical decisions measured? Were they based on actual doctor actions or only predictions?

**Strengths:**

Very important topic: Using foundation models in medical image tasks is new and helpful for hospitals.
Good mix of data: The paper uses both public datasets and real data from clinics, which makes the results more realistic.
Covers many tasks: The authors test both segmentation and decision-making, not just one thing.
Human evaluation: They ask real doctors to check the AI outputs, which is very valuable for healthcare.
Well-written: The paper is clear, organized, and gives good visual results and tables.

**Summary Of The Paper:**

This study explores the potential of large pre-trained AI models for tasks in medical image segmentation and clinical decision-making. The authors examine how models such as SAM (Segment Anything Model), MedSAM, and DINOV2 performed. It was tested not only on open-access datasets but also on real clinical data collected from hospitals which is very interesting.

They assess these models both in zero-shot scenarios, where the model is used as-is without any additional training and with models that are fine-tuned using domain-specific medical data. The tasks under investigation include segmenting organs, detecting tumors and even more complex ones like predicting whether a surgical intervention might be needed.

Importantly, the study involves radiologists who evaluate the output of the AI models which is very important. This step is meant to ground the findings in actual clinical workflows, providing a clearer sense of whether the models’ predictions are meaningful and actionable in real-world practice.

**Weaknesses:**

Only a few experts were involved just three radiologists gave their feedback, so more opinions would have made the results stronger.
Some of the methods aren’t fully shared. For example, certain models or how they were fine-tuned aren’t easy for others to copy.
There’s also no mention of how much time or computing power these models need, even though some of them can be very demanding.
Finally, most of the focus was on image tasks. The parts about clinical decisions weren’t explained as clearly or in as much detail.

---

### Official Review · Reviewer_Ddyi · 2025-07-16
**Beyond Detection: Comparative Explainability Study on Trypanosoma cruzi Using CAMs and DETR Attention**

**Confidence:** 4
**Clarity Of Writing:** excellent
**Clinical Significance:** excellent
**Methodological Novelty:** great
**Overall Rating:** 7

**Experiments And Results:**

great

**Questions For The Authors:**

Does the model learn features unique to T. Cruzi? How would it perform if there are other parasites in the blood smears?

**Strengths:**

Well-written: introduction clearly motivates the need for more interpretable and transparent models for clinical diagnosis.

Broad CAM benchmarking: evaluates a wide range of CAM variants on YOLOv8 alongside query-specific attention maps from DINO-DETR.

Model comparisons: thorough comparison between CAM-based and DETR attention approaches, clearly contrasting their behaviors and illustrating how they complement each other.

Dual-metric evaluation: uses IoU and EBPG, explains each metric’s strengths and limitations, and demonstrates how they complement each other. The paper also emphasizes the need for a new unified metric that encompasses both.

**Summary Of The Paper:**

Clinically interpretable deep learning models are essential for medical diagnostics to enable clinicians to trust and understand AI-driven decisions. However, detecting parasites like T. cruzi in microscopic blood-smear images remains challenging, partly because existing deep-learning methods lack transparency and interpretability. Therefore, this paper proposes enhancing the explainability of object-detection models, specifically by applying two vision-based approaches: gradient-based CAMs on a fine-tuned YOLOv8 detector and query-specific attention maps from a DINO-DETR model, to clearly identify T. cruzi parasites. The visualization pipeline generates saliency maps, demonstrating that CAMs highlight broader parasite regions while DETR attention focuses on more precise parasite features, as validated using IoU and EBPG metrics.

**Weaknesses:**

While the paper aims to support clinicians, it does not evaluate whether saliency maps actually improve diagnostic accuracy, clinician trust, or offer meaningful localization benefits beyond what is already provided by predicted bounding boxes.

Adding a visual pipeline diagram for the deformable attention saliency extraction (Section III.E) might help improve clarity for readers.

---

### Official Review · Reviewer_KHMt · 2025-07-17
**Beyond Detection: Comparative Explainability Study on Trypanosoma cruzi Using CAMs and DETR Attention**

**Confidence:** 2
**Clarity Of Writing:** good
**Clinical Significance:** good
**Methodological Novelty:** fair
**Overall Rating:** 5

**Experiments And Results:**

good

**Questions For The Authors:**

Good work

**Strengths:**

This paper presents a comprehensive explainability study comparing CNN- and transformer-based object detection models for automated Trypanosoma cruzi detection. The authors introduce a novel query-specific attention visualization method for DINO-DETR, which maps decoder attention weights to spatial regions, enabling fine-grained interpretability without backpropagation. They also perform the first systematic benchmarking of ten gradient-based CAM-XAI methods across multiple internal layers of YOLOv8, identifying the most effective method-layer combinations for visualizing parasite features. Using quantitative localization metrics, this study offers the first side-by-side evaluation of explainability techniques for CNN and transformer architectures in a medical imaging context, highlighting important differences in how each model localizes and interprets visual evidence.

**Summary Of The Paper:**

This study addresses the need for interpretable deep learning methods in detecting Trypanosoma cruzi, the parasite responsible for Chagas disease. While YOLOv8 and DINO-DETR achieve high detection performance on microscopy images, their clinical adoption is limited by low interpretability. To bridge this gap, the authors present the first comparative explainability analysis of CNN- and transformer-based object detectors for T. cruzi detection. They benchmark ten CAM-based XAI methods on YOLOv8 and propose a novel attention visualization technique for DINO-DETR that maps decoder attention to image space without backpropagation. Results show that CAMs capture broader parasite regions, while DETR attention focuses on fine-grained features. The study also highlights the limitations of existing localization metrics in multi-object explainability and emphasizes the need for new evaluation methods tailored to medical imaging.

**Weaknesses:**

A key limitation of this study lies in the inadequacy of current interpretability metrics when applied to dense, multi-object settings with shared saliency maps. Existing metrics fail to capture clinically relevant aspects such as objectwise attention strength, spatial precision, and inter-object attribution fairness. As a result, quantitative assessments may not fully reflect model behavior or trustworthiness in real-world applications. Additionally, reliance on numerical metrics without complementary qualitative visualization can lead to over- or underestimation of a model’s clinical interpretability.